# Continuous and Intermittent Exposure to the Toxigenic Cyanobacterium *Microcystis aeruginosa* Differentially Affects the Survival and Reproduction of *Daphnia curvirostris*

**DOI:** 10.3390/toxins16080360

**Published:** 2024-08-15

**Authors:** Fernando Martínez-Jerónimo, Lizabeth Gonzalez-Trujillo, Miriam Hernández-Zamora

**Affiliations:** Instituto Politécnico Nacional, Escuela Nacional de Ciencias Biológicas, Laboratorio de Hidrobiología Experimental, Carpio y Plan de Ayala s/n, Ciudad de México 11340, Mexico

**Keywords:** harmful algal blooms, cyanotoxins, microcystins, aquatic ecotoxicology, Cladocera

## Abstract

Anthropic eutrophication leads to water quality degradation because it may cause the development of harmful cyanobacterial blooms, affecting aquatic biota and threatening human health. Because in the natural environment zooplankters are exposed continuously or intermittently to cyanotoxins in the water or through cyanobacterial consumption, this study aimed to assess the effects of the toxigenic *Microcystis aeruginosa* VU-5 by different ways of exposure in *Daphnia curvirostris*. The acute toxicity produced by the cells, the aqueous crude extract of cells (ACE), and the cell-free culture medium (CFM) were determined. The effect on the survival and reproduction of *D. curvirostris* under continuous and intermittent exposure was determined during 26 d. The LC_50_ was 407,000 cells mL^−1^; exposure to the ACE and CFM produced mortality lower than 20%. *Daphnia* survivorship and reproduction were significantly reduced. Continuous exposure to *Microcystis* cells caused 100% mortality on the fourth day. Exposure during 4 and 24 h in 48 h cycles produced adult mortality, and reproduction decreased as the exposure time and the *Microcystis* concentrations increased. The higher toxicity of cells than the ACE could mean that the toxin’s absorption is higher in the digestive tract. The temporary exposure to *Microcystis* cells produced irreversible damage despite the recovery periods with microalgae as food. The form and the continuity in exposure to *Microcystis* produced adverse effects, warning about threats to the zooplankton during HCBs.

## 1. Introduction

Freshwater ecosystems are increasingly affected by human activities. Water resource degradation is one of the major problems facing humankind; this situation deteriorates the availability and quality of water for human consumption and use in a multiplicity of productive activities. Runoff from agricultural cultivation areas and treated and untreated wastewater discharges cause anthropogenic eutrophication [1]. Eutrophication is a phase of succession in aquatic communities that is characterized by an increase in primary productivity in a water body as a result of an increase in the nutrients, mainly nitrogen, and phosphorus, which causes the accumulation of organic matter and leads to the degradation of aquatic ecosystems, reducing water quality and affecting the uses of hydric resources [2]. The main effects of anthropogenic eutrophication are increased incidences of harmful microalgal and cyanobacterial blooms (HABs and HCBs), leading to reduced dissolved oxygen concentration; habitat loss with changes in biodiversity; adverse impacts to fisheries, tourism, and sports activities; and risks to animal and human health [3].

Massive occurrences of microalgae in which toxic metabolites are produced and released are referred to as “harmful algal blooms” (HABs), and those dominated by cyanobacteria are “harmful cyanobacterial blooms” (HCBs), and both phenomena represent a significant increase in primary productivity. Cyanobacterial-dominated blooms (HCBs) are the most frequent in freshwater lakes and represent a risk to aquatic biota and freshwater uses due to the ability of some cyanobacteria to produce cyanotoxins [4]. In lakes located in temperate regions in the northern hemisphere, blooms usually occur in late summer (June–July) and develop over periods of hours to days [5]; these may remain until the end of summer. Nevertheless, in subtropical latitudes, blooms can be observed throughout the year and can be permanent [6]. The possibility that a cyanobacterial bloom may be toxic is present in more than 50% of the cases. Also, the toxicity of cyanobacteria may vary each week at the same sampling point. Likewise, mixed high- and low-toxicity zone patterns have been found in the same lake [7]. These blooms are evident by the coloration of the water bodies, generally green or blue-green, with turbidity, reduced transparency, and unpleasant odor (that is conferred to the water), or by the presence of scum accumulations or dense superficial strips near the shore of the lakes [8,9].

*Microcystis aeruginosa* is one of the most important and frequent species causing cyanobacterial blooms in freshwater ecosystems [10,11]. This cyanobacterium presents unicellular, spherical morphology with a diameter of 4–5 µm; however, in cyanobacterial blooms, it can be found organized in amorphous colonies enveloped in amorphous mucilage of variable size (52–200 µm) [12]. It has a cosmopolitan distribution, and some strains produce toxins (microcystins) that accumulate inside the cell and are released into the water when cell lysis occurs or under stress conditions [13]. Microcystin poisoning in humans is caused by the direct ingestion of water containing microcystins, contact during recreational or sporting activities, the inhalation of aerosols, and consuming contaminated food. In the aquatic environment, intoxication occurs by direct exposure of biota to water with toxins during and after HCBs or by the ingestion of cyanobacteria; it is also promoted through trophic relationships, by the predation of organisms exposed to cyanotoxins that may have bioaccumulated cyanotoxins [14,15].

Reports of frequent and permanent cyanobacterial blooms in different bodies of water represent a risk to aquatic biota, water-drinking animals, and humans. Cladocerans are fundamental for the functioning of freshwater systems, as they play an essential role in energy transfer, stability, productivity, and increasing the diversity of trophic webs [16]. Most planktic cladocerans are non-selective filter feeders that consume planktonic microalgae, single-cell cyanobacteria, and detritus [17].

Cladocerans represent the zooplankton in aquatic communities and, when used in ecotoxicological assessment, are applicable models for evaluating the impact of toxic substances on water bodies. This assessment is performed through the controlled exposure of these test organisms. In particular, *Daphnia curvirostris* is a cladoceran considered exotic in Mexico [18] but has shown convenient characteristics as a test organism and even higher sensitivity than *D. magna* [19], so it was selected for our study.

During phytoplankton blooms on the water surface, it is common for cyanobacteria to dominate and displace microalgae [6]. Cyanobacteria dominance is due to various strategies they have, which include their ability to locate at different depths in the water column thanks to aerotopes that confer positive buoyance, their capacity to develop efficiently in low light intensities, and the fact that some species can use atmospheric nitrogen. Added to the above is their ability to synthesize cyanotoxins, and, although the ecological role of these secondary metabolites is not well established at all, these metabolites are known to act as allelopathic substances [11] and cause lethal and acute effects to filter-feeding zooplankton and in general to aquatic biota during and after the senescence of an HCB [20,21]. The adverse effects of HCBs are not only related to the cyanotoxins that some species can synthesize, but other secondary metabolites produced by cyanobacteria can also have biological activity and negatively affect aquatic biota that come into contact with these compounds, either through consumption or in the aquatic environment when they are released [20]. Because HCBs can develop in short times (from a few days to some weeks) and zooplankton can also migrate vertically in the water column, it is essential to determine which cell concentrations can be lethal and which could have chronic effects, taking as variants the cell densities and exposure times of zooplankton to cyanobacteria.

Therefore, the objective of this study was to evaluate the effect of different forms of exposure to a toxigenic strain of *M. aeruginosa* on the survival, reproduction, and body growth of *D. curvirostris*. This study analyzed the effects of continuous and intermittent exposure, with recovery periods of feeding with microalgae. In this study, we tested whether a filter-feeding species, like the cladoceran *D. curvirostris*, can thrive and develop correctly when a toxigenic cyanobacterium is supplied as food. If not, whether time exposure to this cyanobacterium, with alternating recovering time in a healthy environment, can reverse the possible impairing effects was tested. Also, we determined if cyanobacterial crude extracts or the medium in which the cyanobacteria developed has similar toxic effects in the cladoceran. This is to recreate a simile to possible natural conditions when HCBs occur sporadically or periodically and how different ways of exposure to toxigenic cyanobacteria affect this cladoceran.

## 2. Materials and Methods

### 2.1. Test Organisms

The strains of all the test organisms used in this study were obtained from the Microalgae and Cladocerans Collections of the Laboratorio de Hidrobiología Experimental, Escuela Nacional de Ciencias Biológicas, Instituto Politécnico Nacional.

The cladoceran *Daphnia curvirostris* was isolated from samples collected in the Chimaliapan wetland (State of Mexico, Mexico) in February 2008. Parthenogenetic females were isolated and cultured in reconstituted hard water [22] and fed with cultured green microalgae. The clonal strain was obtained by isolating neonates from one single parthenogenetic female. This clonal strain has been successfully maintained in controlled cultures of known age in 500 mL vessels, with 400 mL of a culture medium, supplying the Chlorophycean microalga *Pseudokirchneriella subcapitata* (1 × 10^6^ cell mL^−1^), in an environmental chamber, at 25 °C, and photoperiod 16:8 (light/darkness). All the progeny was separated and discarded daily in these maintenance cultures to avoid crowding and waste accumulation. The culture medium and food were wholly renewed once a week. These stock cultures were replaced every 21 days, starting with new neonates, to always know the age of individuals. We obtained the test organisms from these stock cultures to start all the experiments, separating neonates from the third reproduction. Autoclaved Bold’s Basal Medium was used for *P. subcapitata* propagation.

The toxigenic strain of *Microcystis aeruginosa* VU-5 was used for acute and chronic toxicity assays. This strain was isolated from phytoplankton samples collected in the Olympic Rowing and Canoeing Track “Virgilio Uribe” (VU), Xochimilco, Mexico City, in 2009; the toxigenicity of this strain was genetically determined through the identification of the *mcy* gene cluster [23]. It was grown in 1000 mL Erlenmeyer flasks with 800 mL of a Z8 mineral medium [24], incubating for 15 days at 25 °C, and the illumination of 83 µMol m^−2^ s^−1^. Under the conditions described, this cyanobacterium grows as single cells that do not cluster into colonies. *M. aeruginosa* cells were obtained from 200 mL of the cultures described above.

The crude aqueous extract (CAE) was obtained by the cell disruption of the separated biomass by centrifugation at 4500 rpm for 10 min from 75 mL of the *M. aeruginosa* culture. For disruption, the cell package was exposed to alternating cycles of freezing (−70 °C) and heating in a thermoblock (40 °C), maceration with dry ice, and solubilization in reconstituted hard water, with subsequent filtration to remove cell debris.

On the other hand, to evaluate the presence of extracellular cyanotoxins, a cell-free culture medium (CFM) was obtained by centrifuging 75 mL of *M. aeruginosa* culture at 4500 rpm for 10 min; the biomass was discarded, and the supernatant constituted the CFM.

### 2.2. Acute Toxicity Bioassays with D. curvirostris

The acute toxic effect (48 h) of *M. aeruginosa* was determined for the following cell concentrations: 1.5, 3, 6, 12, and 18 × 10^5^ cells mL^−1^, and the equivalent of these cell densities in the CAE and the CFM. Bioassays were performed according to the Organisation for Economic Co-operation and Development (OECD) test method 202 [25], using neonates (age < 24 h) of *D. curvirostris*. For each exposure condition, three bioassays were performed. Cell densities were determined through counts in the Neubauer chamber using a microscope.

### 2.3. Chronic Evaluation of the Effect of M. aeruginosa Cells on D. curvirostris

#### 2.3.1. Continuous Exposure Bioassays

To evaluate the effect of *M. aeruginosa* consumption and its potential toxicity on the survival and reproduction of *D. curvirostris*, three concentrations (4, 8, and 12 × 10⁵ cells mL^−1^) were tested and contrasted with a control series fed with 1 × 10^6^ cells mL^−1^ of *P. subcapitata*. In all cases, there were 10 replicates containing one neonate each. Assays were performed in 100 mL glass vessels containing an 80 mL exposure medium. Total replacement of the culture medium plus the respective concentration of the microalgae or *M. aeruginosa* was performed every 24 h for a maximum period of 26 days. The experiments were incubated at 25 °C, with a 16:8 photoperiod in an environmental chamber. The test vessels were checked daily to record mortality and reproduction.

#### 2.3.2. Intermittent Exposure Bioassays (24 h)

For this experiment, the organisms were exposed to *M. aeruginosa* cells for 24 h, alternating with 24 h recovery periods. Based on the results of the previous bioassay, four concentrations of *M. aeruginosa* (0.5, 1, 2, and 4 × 10^5^ cells mL^−1^) were used for this assay, and two control series were included. The first control series (C1) consisted of organisms fed with 1 × 10^6^ cells mL^−1^ of the microalgae *P. subcapitata*. The second control group (C2) was implemented to rule out that the possible absence of feeding due to the lack of consumption and utilization of *M. aeruginosa* as food for the cladoceran could be the cause of the observed effects, so for this control series (C2), the organisms were kept fasting during the periods of exposure to *M. aeruginosa*. There were 10 replicates in all cases, consisting of 100 mL beakers with 80 mL of the exposure medium. The experiments were initiated by placing a neonate in each replicate.

This assay consisted of feeding *P. subcapitata* (1 × 10^6^ cells mL^−1^) to all organisms of the four treatments and controls in the first 24 h. After this time, the medium was changed, transferring control 1 to the medium with 1 × 10^6^ cells mL^−1^ with *P. subcapitata*. After this time, the medium was changed entirely, transferring control 1 to the medium with 1 × 10^6^ cells mL^−1^ with *P. subcapitata*. Control 2 was placed in a reconstituted hard water medium without cyanobacteria or microalgae, and the treatments with cyanobacteria were exposed to the four indicated concentrations of *M. aeruginosa*; this condition was maintained again for 24 h. At the end of this time, all treatments and controls were transferred again to a fresh medium, but now all were supplied with 1 × 10^6^ cells mL^−1^ of *P. subcapitata*. In the case of the organisms exposed to *M. aeruginosa*, several washes were made in a fresh dilution medium to avoid contamination with the cyanobacteria in the recovery condition. This procedure was repeated during the 26 days of the experiment. Incubation conditions were similar to those applied in the continuous exposure test. The test containers were checked daily, and from the first reproduction, the neonates were removed and counted. Surviving adult organisms at the end of the tests were fixed in 10% formalin to measure body dimensions. Total length (TL, measured from the top of the head to the extreme of the caudal spine), body length (BL, the length from the top of the head to the insertion of the caudal spine), and maximum body width (MW, the maximum perpendicular length from the dorsal to ventral region) were determined using a stereomicroscope and the software Olympus^®^ cellSens Standard ver. 1.0.

#### 2.3.3. Intermittent Exposure Bioassays (4 h)

Four concentrations of *M. aeruginosa* (0.2, 0.4, 0.8, and 1.6 × 10^5^ cells mL^−1^) were used for this assay, which, based on the results previously obtained, were slightly lower than those previously evaluated. In this case, only a single control series was settled. For each concentration, 10 replicates (100 mL beakers with 80 mL of exposure medium) were set up, with one neonate in each. The assay consisted of feeding 1 × 10^6^ cells mL^−1^ of the microalga *P. subcapitata* to the organisms of the four treatments and the control in the first 44 h. Then, for the next 4 h, the organisms of the four treatments were exposed to the mentioned amounts of *M. aeruginosa*, and the control was maintained with 1 × 10^6^ cells mL^−1^ of *P. subcapitata*. Each time the organisms were changed from microalgae to *M. aeruginosa*, careful washes were made in reconstituted hard water to avoid dragging cells from the cyanobacteria exposure medium; this procedure was repeated during the 26 days of the experiment. Incubation conditions were 25 °C and a 16:8 photoperiod. The test containers were checked daily, and from the first reproduction, the neonates were removed and counted. At the end of the assays, the surviving adult organisms were fixed in 10% formalin to measure the total length (TL), body length (BL), and maximum body width (MW) as described previously and evaluate possible effects on body dimensions.

### 2.4. Quantification of Microcystins

The microcystin concentration in *M. aeruginosa* cells, in the crude aqueous extract (CAE), in the cell-free culture medium (CFM), and in the different concentrations of *M. aeruginosa* used in the acute toxicity bioassay was determined with the QuantiPlate™ Kit for Microcystins from Envirologix^®^, which is an ELISA (Enzyme-Linked Immunosorbent Assay) evaluation. This kit allows for the quantitative detection of four microcystins (MC-LR, MC-LA, MC-RR, and MC-YR) in addition to nodularin in a range of 0.1 to 2.5 µg L^−1^ (the Limit of Detection (LOD) is 0.10 µg L^−1^). However, the result is expressed only as MC-LR equivalents.

### 2.5. Statistical Analysis

The mean lethal concentration (LC_50_) of *D. curvirostris* in acute toxicity bioassays (48 h) for cell concentration, ACE, and CFM of *M. aeruginosa* was determined with the Probit method using Risk Assessment (RA) software ver. 1.0.

Survival in chronic assays was analyzed using the Long-rank Mantel–Cox test, and significant differences were established with the Holm–Sidak test. Reproductive responses (total progeny, age at first reproduction, and number of clutches) were analyzed by a one-way analysis of variance (ANOVA-I). Where significant differences were found, post hoc Tukey’s multiple comparisons and Dunnett’s test for comparison with the control were performed. Statistica ver. 14 and GraphPad Prism 10 software were used in all cases.

## 3. Results

### 3.1. Acute Toxic Effects of M. aeruginosa

The average LC_50_ of three bioassays for exposure to cyanobacteria cell concentration was 4.07 × 10^5^ cells mL^−1^ (95% confidence limits: 3.04–5.13 cells mL^−1^); this value refers to the cell density of *M. aeruginosa* that produced 50% mortality in *D. curvirostris* neonates after 48 exposure, indicating that the presence and possible consumption of cyanobacteria produced lethal effects that were proportional to the cell concentration.

In the case of exposure to the *M. aeruginosa* cell-free culture medium, mortality was only recorded at the equivalent of the two highest concentrations of the cyanobacteria (2 and 4% on average for the CFM equivalent to 12 and 18 × 10^5^ cells mL^−1^). This reduced mortality demonstrates that the cyanotoxins were released minimally to the culture medium. This result also helps to understand that the mortality response recorded in the acute bioassays with the *M. aeruginosa* cells was because the neonates consumed them and not because of extracellular toxins.

For the acute toxic response to the aqueous crude extract (ACE), increasing mortality was recorded in proportion to the equivalent cell density of *M. aeruginosa*. However, the maximum value recorded was 20% for the concentration of 18 × 10^5^ cells mL^−1^, so it was not possible to calculate the LC_50_.

### 3.2. Survival Effects of Chronic Exposure to M. aeruginosa

#### 3.2.1. Continuous Exposure

It was initially planned for this experiment to evaluate for 26 d the effect of continuous exposure to the toxigenic *M. aeruginosa* provided as food to *D. curvirostris*, but 100% mortality was recorded on day 2 for the dose of 12 × 10^5^ cells mL^−1^, and on the fourth day for the concentrations of 4 and 8 × 10^5^ cells mL^−1^, so this assay was suspended after four days. This lethal response is evidence of the high toxicity of this strain on *D. curvirostris*. The survivorship of organisms fed with *P. subcapitata* was 100%.

#### 3.2.2. Intermittent Exposure

Figure 1 shows the survival of *D. curvirostris* alternately exposed for 24 h to *M. aeruginosa* with 24 h of recovery for 26 days. This figure shows a trend of increasing mortality in adult females as the concentration of cyanobacteria increases. Partial mortality was also recorded in control 2 (maximum of 30%), which can be explained by the 24 h starvation period these females had. However, the Long-rank Mantel–Cox test established no significant differences in survival between the two series controls. All the survival curves recorded in the *M. aeruginosa* treatments were significantly different from those in both control series. The Holm–Sidak test established that survivorship was lower at all concentrations of *M. aeruginosa*; nevertheless, the values recorded for the three highest concentrations did not differ significantly (*p* > 0.05, Figure 1). Exposure to the maximum concentration of *M. aeruginosa* produced 100% mortality in *D. curvirostris* females at day 18. In contrast, for the remaining concentrations of cyanobacteria, all females died between days 25 and 26, reflecting the toxic effect produced by the consumption of the toxigenic cyanobacteria and the release of cyanotoxins during the feeding of the cladocerans.

Figure 2 shows the survivorship curves of *D. curvirostris* for the short exposure of 4 h with 44 h of recovery. Even though *D. curvirostris* had a reduced time of consumption and exposure to microcystins, there were adverse effects on the survival of the females that were proportional to the concentration of *M. aeruginosa*. However, there was no 100% mortality during the 26 d duration of the tests. The Holm–Sidak test showed that there was no difference between the survival of the control and the two lower concentrations of *M. aeruginosa* (0.5 and 1 × 10^5^ cells mL^−1^), but the two higher doses of this cyanobacteria significantly reduced the survival of *D. curvirostris*.

### 3.3. Reproductive Effects of Chronic Exposure to M. aeruginosa

The reproductive responses of *D. curvirostris* exposed to *M. aeruginosa* for 24 h, with 24 h recovery, are shown in Figure 3. The ANOVA for the accumulated total progeny was significant (F = 50.87, *p* < 0.001). Figure 3 shows that all cell concentrations of *M. aeruginosa* significantly affected fecundity and reduced the total progeny of *D. curvirostris* (Dunnett’s test, *p* < 0.5), proportionally to the increase in the cyanobacteria density, but also, total progeny in C2 was significantly lower than that recorded in C1. Significant reduction in the total progeny in control 2 indicates that the 24 h starvation affected fecundity but not to the extent that cyanobacteria consumption did. There was no significant effect of *M. aeruginosa* concentration on the age of first reproduction (ANOVA F = 0.311, *p* = 0.904). The number of clutches was significantly reduced at all concentrations of *M. aeruginosa* (ANOVA F: 49.639, *p* < 0.001; Dunnett’s test, *p* < 0.05); in this case, there was no difference between the controls C1 and C2. Figure 3 also shows the results of Tukey’s pairwise comparison test, indicating the differences among all the treatments, including the controls; according to this test, total progeny was significantly reduced in all the cyanobacterium concentrations, but no differences were detected among them. Furthermore, the number of released clutches was the lowest in the three highest *M. aeruginosa* concentrations but was not different among them.

Figure 4 shows the reproductive responses of *D. curvirostris* intermittently exposed to different concentrations of *M. aeruginosa* for 4 h, with 44 h of recovery, fed with *P. subcapitata*. ANOVA applied to the total progeny demonstrated significant differences (*p* = 12.967, *p* < 0.001). Even though lower quantities of the cyanobacterium were fed in this experiment, total progeny was significantly reduced in proportion to cell density in all the treatments, according to Dunnett’s test (*p* = 0.05, Figure 4). However, age at first reproduction was not affected (ANOVA F = 0.529, *p* = 0.715). Regarding the number of clutches released during the 26 days of the experiments, the applied ANOVA evidenced significant differences (F = 12.967, *p* < 0.001), and, according to Dunnett’s test, clutches were significantly lower at all the *M. aeruginosa* concentrations higher than 0.4 × 10^5^ cells mL^−1^. The Tukey’s test indicated that the clutches were only lower in the two highest *M. aeruginosa* concentrations and that there were differences between them (Figure 4).

### 3.4. Effects on Body Size of Chronic Exposure to M. aeruginosa

Figure 5 shows the total length (TL), body length (BL), and maximum body width (MW) of *D*. *curvirostris* females exposed to *M. aeruginosa* for 24 h, with alternating periods of recovery (24 h), for 26 days. The ANOVA for TL evidenced significant differences (F = 2.929, *p* = 0.021), and also differences were determined for the BL (F = 2.532, *p* = 0.0.048) but not for the MW (F = 0.247, *p* = 0.939). There was no difference in any of the measurements for C1 and C2. However, it is notable that organisms exposed to cyanobacteria significantly reduced TL at all concentrations (*p* = 0.05), while body length was lower at concentrations of 2 and 4 × 10^5^ cells mL^−1^ (*p* = 0.05). Body width was not different in any case.

Figure 5 also shows the size of females exposed for 4 h to *M. aeruginosa* with 44 h of recovery. The ANOVA demonstrated significant differences for TL (F = 2.578, *p* = 0.046), and for BL (F = 2.532, *p* = 0.0.048), but not for the MW (F = 0.185, *p* = 0.939). In this case, the organisms only had lower TL and BL at the concentration of 1.6 × 10^5^ cells mL^−1^, and at the concentration of 0.8 × 10^5^ cells mL^−1^, only the BL was lower than the control. There were no significant differences in the maximum body width.

### 3.5. Microcystin Quantification

The concentration of microcystins determined by the ELISA method (QuantiPlate™ Kit for Microcystins) showed that in the cell-free culture medium (CFM), no microcystins were recorded (LOD = 0.1 μg L^−1^), which allows for asserting that the toxins were mainly present inside the cells. In the crude aqueous extract, which through rupture allowed for the release of the entire intracellular content, including cyanotoxins, the concentration was 2.1 μg L^−1^, while for the cells, it was 1.6 μg L^−1^. It should be noted that the quantification of microcystins was not performed for the different concentrations of the acute bioassays of ACE and CFM of *M. aeruginosa* because the mortality was less than 20%.

In the acute bioassay with *M. aeruginosa* cells, the concentration of microcystins was quantified at all cell densities tested; these values are shown in Figure 6. A proportional relationship between concentration and cell density of the cyanobacteria is observed, which coincided with the toxic effects observed in these acute assays.

## 4. Discussion

Exposure of *D. curvirostris* to cells of the toxigenic strain of *M. aeruginosa* simulates what might occur when zooplankton are exposed to a harmful cyanobacterial bloom. During a bloom, the filter-feeding organisms might consume these cells or be exposed to toxins released extracellularly or during cell disruption.

The acute toxicity bioassay for direct exposure of *D. curvirostris* to *M. aeruginosa* cells determined that the LC_50_ ranged from 368,000 to 449,000 cells mL^−1^, with an average of 407,000 cells mL^−1^; this is a lower value than that obtained by the authors of [26], who report an LC_50_ of 1.752 × 10^7^ cells mL^−1^ for *D. magna* with *M. aeruginosa*. This difference can be explained by the different capacities of the toxigenic strains of this cyanobacterium to synthesize cyanotoxins. However, environmental factors that promote the expression of the genes responsible for this synthesis also play a crucial role, which could lead to toxin production variation even in the same strain.

In our acute toxicity study, the highest concentration of *M. aeruginosa* produced almost 100% mortality. As shown in Figure 6, the concentration of the toxins quantified was related to the cell density to which *D. curvirostris* neonates were exposed. The observed lethal response demonstrates that the toxic effects were produced by intracellular toxins, which were released into the digestive tract by the consumption of the cells. This is confirmed by the lack of effects that the CFM produced and the less than 20% toxicity provoked by the ACE. As has been documented, microcystins are located 90% inside cyanobacterial cells, where they could have some important functions for the cell, such as iron metabolism and light adaptation [27,28].

However, it has also been reported that cyanotoxins could be released extracellularly via transporter proteins and have allelopathic or interacting functions with other components of the planktonic community [27,29]. These secondary metabolites may also be released under chemical or environmental stress [30,31] and not always after the cell lysis that occurs subsequently to the senescence of a cyanobacterial bloom [2]. In our study, no extracellular toxins were recorded, which could indicate that none of the conditions for toxins to be released to the medium were present in the propagation cultures.

On the other hand, the lower lethal effect recorded in the crude extract (less than 20%) in equivalent exposures at the same cell concentrations can be explained by the route of exposure of *D. curvirostris*, which was not primarily related to external contact to the cyanobacterial cell contents, but rather the effect occurred through the release of the toxins in the digestive tract after the consumption of the cells. The lack of effect of CFM and the lesser lethal effect of ACE are similar to that reported by Olvera-Ramírez et al. [16], who evaluated the toxic effects of *Pseudanabaena tenuis* on the cladocerans *D. magna* and *Ceriodaphnia dubia*.

The acute toxicity of crude aqueous extracts of *M. aeruginosa* has been reported in other studies, recording lethal effects of both toxigenic strains and non-toxigenic cyanobacteria. For example, Pineda-Mendoza et al. [32] report an LC_50_ for *D. magna* neonates that ranged from 363.91 to 741.1 mg L^−1^, as biomass in dry weight; these are very high values and cannot be compared with what was evaluated here, since in our case we used exposure to different cell densities and their equivalent in the aqueous crude extract. However, another study [33] reports that the toxicity of extracts was higher than that produced by pure microcystin-LR, which they indicate is due to the presence of other secondary metabolites with biological activity that are released upon cell disruption and increase the toxicity of cyanotoxins. This situation did not occur in our study since the most significant toxic effects were recorded in the exposure to *M. aeruginosa* cells and not in the cellular equivalent of aqueous crude extracts.

Microcystins are monocyclic heptapeptides that different cyanobacteria can synthesize. Currently, it is known that there are more than 200 different types of microcystins that present structural variants in their molecule, such as variations of amino acid positions, which confers them different physicochemical and toxic properties [34]. In our study, four microcystins (microcystin-LR, -LA, -RR, -YR) were quantified, in addition to nodularin, which are the cyanotoxins detected by the Quantiplate^®^ kit. However, it is not excluded that other microcystins may have been present in the cells and the ACE and were not detected by the analysis used. In any case, *D. curvirostris* could integrate all the impairing effects of the metabolites with biological activity and express these affectations through the lethal and sublethal responses evaluated here.

On the other hand, it has also been documented that microcystins are not the only secondary metabolites with biological activity produced by cyanobacteria. Among others, dihydroxy pyrrolidone (which is a neurotoxin that inhibits digestive glycosidases in crustaceans), microviridins, aeruginosins, cyanopectolins, anabaenopeptins, and cyclamides are also cyclic peptides that have been documented in crude extracts of cyanobacteria [14,23,32,35], most of these producing the inhibition of protease activity [36]. Although most of these metabolites are considered non-toxic, their biological activity in some cases has been superior to that of purified cyanotoxins, in addition to the fact that the adverse effects of crude extracts also depend on the cyanobacterial species [33].

The results obtained here demonstrate that the toxic effect of the crude extract at concentrations equivalent to the cell density of *M. aeruginosa* was lower than that of the whole cells, which could perhaps be due to possible antagonistic effects between some molecules of the intracellular content and that the ACE content could not enter the organism as it occurs with the cells filtered and concentrated by neonates. Remarkably, the concentration of the cells by filtration and the lysis process in the digestive tract would mark a difference with the ACE, which could not be concentrated and could not have direct access to the digestive tract of the test organisms.

In the chronic assays, the high toxicity of the *M. aeruginosa* strain as a potential food source for *D. curvirostris* was documented in the initial assay with 4, 8, and 12 × 10⁵ cells mL^−1^, where 100% mortality was recorded at day 4 of the beginning of the experiments. Similar results were reported by Chen and Xie [37] in *Daphnia carinata* when they supplied *M. aeruginosa* at concentrations of 1 and 5 × 10^5^ cells mL^−1^ and they also documented 100% mortality on day four. This response is because, in the absence of other available food, the consumption of *M. aeruginosa* VU5 cells provoked toxic effects that drastically reduced the survival of the cladoceran. However, the low nutritional quality documented in cyanobacteria must be added [10]. Our results are similar to those reported by Liu et al. [26], who documented a significant toxic effect of *M. aeruginosa* that reduced *D. magna* survival to 13.3 and 16.7% at 96 h when they supplied *M. aeruginosa* concentrations of 0.75 and 1.5 × 10^7^ cells mL^−1^, respectively. In the study by Liu et al. [26], the control group was fed the microalga *Chlorella* sp., which produced 98.4% survival, while the toxic effects were recorded with *M. aeruginosa* concentrations higher than those tested by us (up to 1.2 × 10^6^ cells mL^−1^); the differences in lethality could be related both to the difference in toxigenicity of the cyanobacterial strains and the cladoceran species in each case.

For the exposure of different times to *M. aeruginosa* cells, with alternating periods of recovery with microalgae as food, it was demonstrated that even exposures as short as 4 h affected the survival of *D. curvirostris*, despite having 44 h of recovery with food similar to that of the control and without contact with the cyanobacteria. The negative effects on survival increased when exposure was augmented to 24 h (with 24 h recovery). The use of *M. aeruginosa* as a diet for cladocerans reduced reproduction and growth and affected their life cycle [33]. It has also been reported that zooplankters that ingest toxigenic cyanobacteria accumulate the toxins by uptake at the digestive tract level [38] and that the filtration rate is inhibited [39,40], which would help to understand why, in addition to all the metabolic and physiological alterations, the organisms could have diminished their ability to feed adequately during recovery periods, with the consequent effects on development and survival. All of the above effects were evidenced by the results we obtained.

One of the ways in which microcystins act is through the inhibition of protein phosphatases 1 and 2A (PP1 and PP2A), which are the most abundant serine/threonine phosphatases in eukaryotic cells and play a key role in the regulation of several cellular functions, ranging from intermediary metabolism to apoptosis. These enzymes are involved in the dephosphorylation of phosphoproteins at serine and threonine residues in the cell cytosol. The inhibition of PP1 and PP2A would leave individuals exposed to microcystins through the diet with an altered physiological condition from which they would not be able to recover. It should be added that exposure to microcystins produces oxidative stress by forming reactive oxygen species (ROS) and inhibits the enzymes trypsin and chymotrypsin, which are responsible for proteolysis. Thus, in addition to direct damage, organisms consume additional energy to maintain several physiological processes, such as the biotransformation of chemical stressors and the synthesis of antioxidant enzymes, which aid in eliminating toxins and cellular repair mechanisms [40].

The total progeny of *D. curvirostris* was reduced when exposed at two different time intervals to *M. aeruginosa*. This may be because cyanobacteria possess a low concentration of lipids and lack sterols and long-chain polyunsaturated fatty acids (PUFAs) [41,42], including eicosapentaenoic acid, which are molecules that affect and limit the growth, reproduction, molting process, and lipid reserves in cladocerans [43,44]. The organisms that covered their diet by consuming *Microcystis* cells, alternating with microalgae (during recovery periods), were subjected to physiological stress, provoked by secondary metabolites such as microcystins, and also nutritional stress due to the deficiency in the composition of *M. aeruginosa*, which finally affected their fecundity and survival.

This chemical stress implies that the organisms had to redistribute their energy, reducing resources for reproduction, as can be seen in the reduction in fecundity in organisms exposed to *M. aeruginosa*. Our results are comparable with those obtained by Olvera-Ramírez et al. [16], who administered feeding continuously for 23 days with the cyanobacterium *Pseudanabaena. tenuis* at dry weight concentrations of 2, 4, and 8 mg L^−1^ and reported a reduction of more than 80% in total progeny in *C. dubia*, being, according to the authors, a more sensitive cladoceran than *D. magna*. In our case, the most significant reduction in the total progeny of *D. curvirostris* was recorded in organisms fed intermittently for 24 h with cell concentrations of 0.5 to 4 × 10^5^ cells mL^−1^ of *M. aeruginosa*, indicating a greater sensitivity of *D. curvirostris* and most probably a greater toxigenic capacity of the strain of *M. aeruginosa* that we used for the studies.

Although the age of first reproduction was delayed in organisms fed intermittently with *M. aeruginosa*, no significant differences were demonstrated in this reproductive parameter, which implies that although there was no delay in gonad maturation, the adverse effects were manifested in the reproductive frequency, the number of clutches produced, and the number of offspring per clutch, which evidenced the toxic effects on fecundity produced by the consumption of this cyanobacteria as food. The effects on reproduction were more acute in the organisms that were exposed for a longer time to *M. aeruginosa* (24 h), which reflects that the negative effect was caused by the toxic metabolites of this cyanobacteria during their consumption as food and by nutritional deficiency, although not necessarily by starvation, as can be inferred from control two, which was without food supply during the times equivalent to the most extended duration of exposure in the treatments with *M. aeruginosa*.

Exposure to cyanobacteria also affected the body growth of *D. curvirostris*. This was manifested as the decrease in length recorded at all concentrations of cyanobacteria in females exposed for 24 h and at the two highest concentrations in the 4 h exposure. Again, stress conditions related to the effects of cyanotoxins, and the low nutritional quality of *M. aeruginosa* led to impaired somatic development in females. Effects of cyanobacteria consumption as food on cladoceran size reduction have been documented by Abrantes et al. [45] who report reduced size at the onset of reproduction and in general also in ovigerous females of *Ceriodaphnia pulchella*, together with reduced fecundity, when phytoplankton diversity, consumed as food by filter-feeding zooplankton, shifted towards cyanobacteria dominance. Similar effects were reported by Olvera-Ramirez et al. [16] in *D. magna* fed with *Pseudanabaena tenuis* for 23 days and also by Liu et al. [26] in *D. magna* supplied with *M. aeruginosa* (1.5, 3.0, and 7.5 × 10^6^ cells mL^−1^) for 13 days; in both cases, the control was fed with microalgae.

The MW did not show significant differences in any concentration of *M. aeruginosa* with respect to the control fed with the microalgae *P. subcapitata*, which means that the toxic metabolites of the cyanobacteria did not affect the decrease in this body measurement.

Finally, regarding the quantification of microcystins in this study, the ACE did contain microcystins at a concentration of 2.1 μg L^−1^ as MC-LR and the produced lethal effects on *D. curvirostris* neonates were relatively low compared to those documented when organisms were exposed to *Microcystis* cells, whose toxin content at the cellular equivalent of the concentration in the ACE was 1.6 μg L^−1^. The quantification procedure we used (ELISA) is designed for water samples. Still, we have used this in suspensions of whole cyanobacteria cells and in determining toxin content in aqueous crude extracts with repeatable results. The differences in the microcystin concentration between ACE and the equivalent number of cells can be explained as follows: during cell disruption, all the aqueous cell content is released and stays in the solution. In contrast, when we process the whole *Microcystis* cells, it is possible that no complete “extraction” of the detectable cyanotoxins could happen. According to this, it is feasible that we could underestimate the real toxin concentration in the entire cell determinations. This issue requires further studies and comparisons when ELISA kits are used to quantify microcystins in different conditions of the samples.

Exposure of *D. curvirostris* to the toxigenic strain of *M. aeruginosa* VU5 caused toxic responses that ranged from lethal to chronic effects in reproduction, survival, and body growth. Cell concentrations that might not be considered so high (4 × 10^5^ cells mL^−1^), which are quite feasible to register during an HCB, provoked 100% mortality over four days. The exposure, alternated with periods of recovery, affected the reproduction and survival of the cladoceran, an effect that is explained as a toxic response to the intracellular microcystins captured through the digestive tract of the organisms once the cells were concentrated by the filtration activity of the thoracopods, ingested and subjected to the digestion process, with the release of the intracellular content to the interior of the individuals; this would also help to explain the lower lethal effect produced by ACE. The exposure time to *M. aeruginosa* affected the development and survival of *D. curvirostris*, even in the short exposure duration (4 h), even though they had complementary recovery times. The above warns about feasible scenarios that could occur in the zooplankton community in the natural environment when HCBs develop, which could affect the structure and diversity of the community by displacing sensitive species or reducing their demographic parameters.

## 5. Conclusions

Temporary, even short-term exposure to *M. aeruginosa* cells caused irreversible damage in *D. curvirostris* despite recovery periods in a fresh culture medium and feeding with microalgae. The consumption of *M. aeruginosa* cells produced a more significant lethal effect on *D. curvirostris than* the aqueous crude extract and cell-free culture medium; this result indicates that the main negative effect of cyanotoxins is through consumption and uptake in the digestive tract. When *D. curvirostris* was continuously exposed to *M. aeruginosa* (4 × 10^5^ cells mL^−1^)*,* the maximum survival was only four days. Cyanotoxins reduced survival and negatively affected cladoceran reproduction, reducing fecundity. In addition, body growth was also affected, registering smaller sizes in females exposed to *M. aeruginosa*; the longer the exposure, the greater the effect. Results allow us to question whether assays using crude aqueous extracts of cyanobacteria adequately reflect their toxigenic potential, given that the most significant adverse effects are generated by the direct consumption of cyanobacterial cells. The results of the present study also indicate that the form and time of exposure of *D. curvirostris* to the toxigenic cyanobacterium *M. aeruginosa* VU-5 produced negative responses of different magnitude, alerting about possible scenarios faced by filter-feeding zooplankton during harmful cyanobacterial blooms.

## Figures and Tables

**Figure 1 toxins-16-00360-f001:**
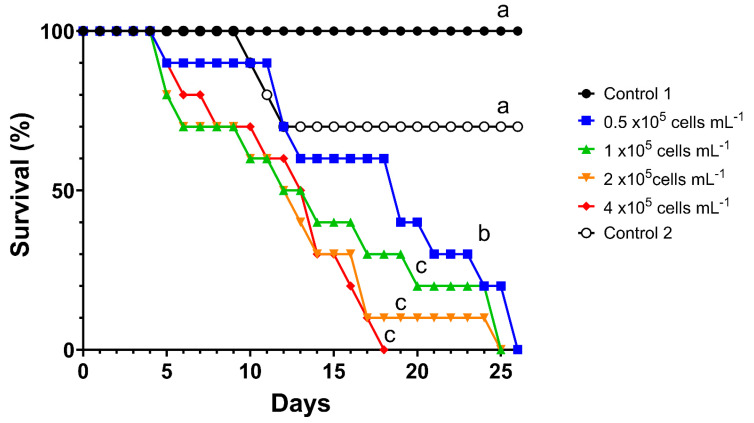
Survival of *Daphnia curvirostris* in alternating 24 h exposure to *Microcystis aeruginosa* with 24 h recovery fed on *Pseudokirchneriella subcapitata*. Different letters indicate significant differences according to the Holm–Sidak test, *p* < 0.05.

**Figure 2 toxins-16-00360-f002:**
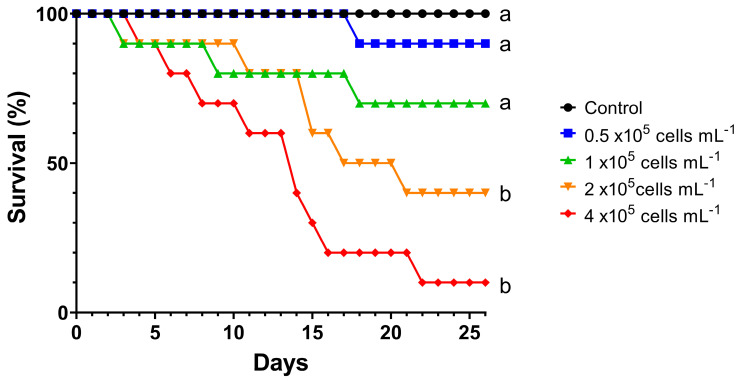
Survival of *Daphnia curvirostris* in alternating 4 h exposure to *Microcystis aeruginosa* with 44 h recovery fed on *Pseudokirchneriella subcapitata*. Different letters indicate significant differences according to the Holm–Sidak test, *p* < 0.05.

**Figure 3 toxins-16-00360-f003:**
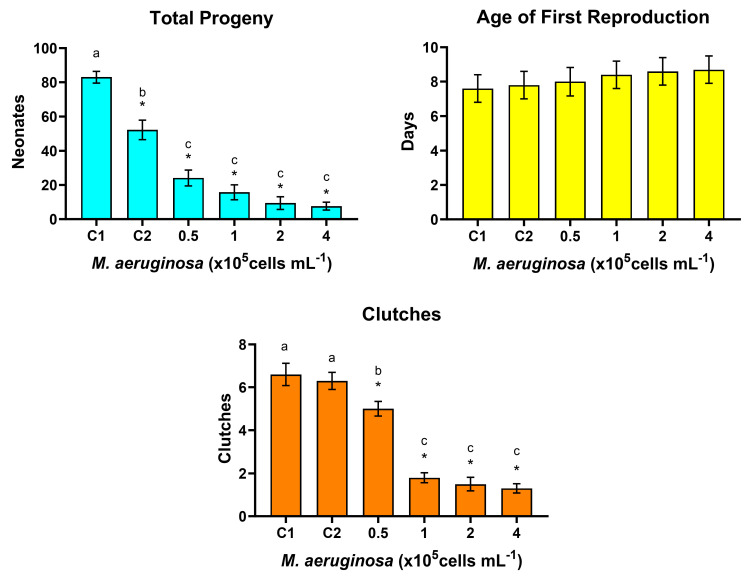
Reproductive responses of *Daphnia curvirostris* in alternating 24 h exposure to *Microcystis aeruginosa* with 24 h recovery fed on *Pseudokirchneriella subcapitata*. Bars are for average values ± standard error limits. Asterisks indicate significant differences from the control (Dunnett’s test, *p* < 0.05); different letters above the columns are for treatments differing significantly (pairwise Tukey’s test, *p* < 0.05).

**Figure 4 toxins-16-00360-f004:**
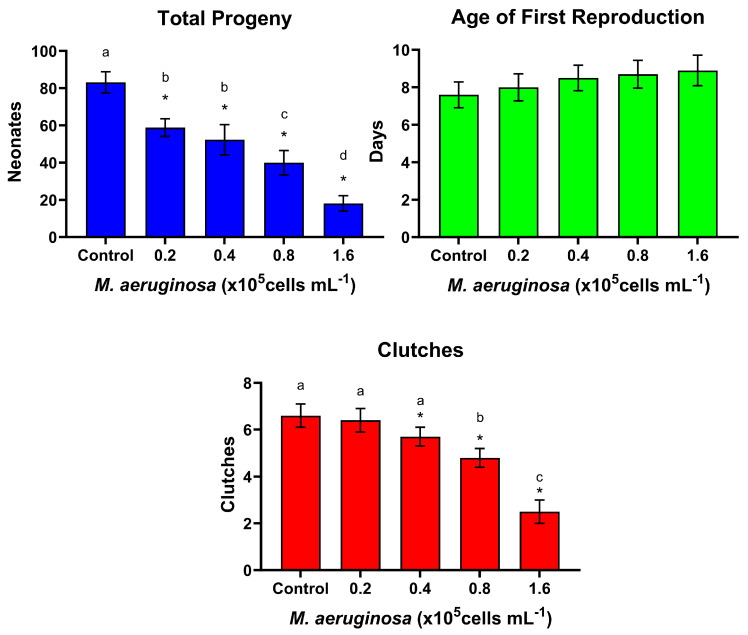
Reproductive responses of *Daphnia curvirostris* in alternating 4 h exposure to *Microcystis aeruginosa* with 44 h recovery fed on *Pseudokirchneriella subcapitata*. Bars are for average values ± standard error limits. Asterisks indicate significant differences from the control (Dunnett’s test, *p* < 0.05). Different letters above the columns are for treatments differing significantly (pairwise Tukey’s test, *p* < 0.05).

**Figure 5 toxins-16-00360-f005:**
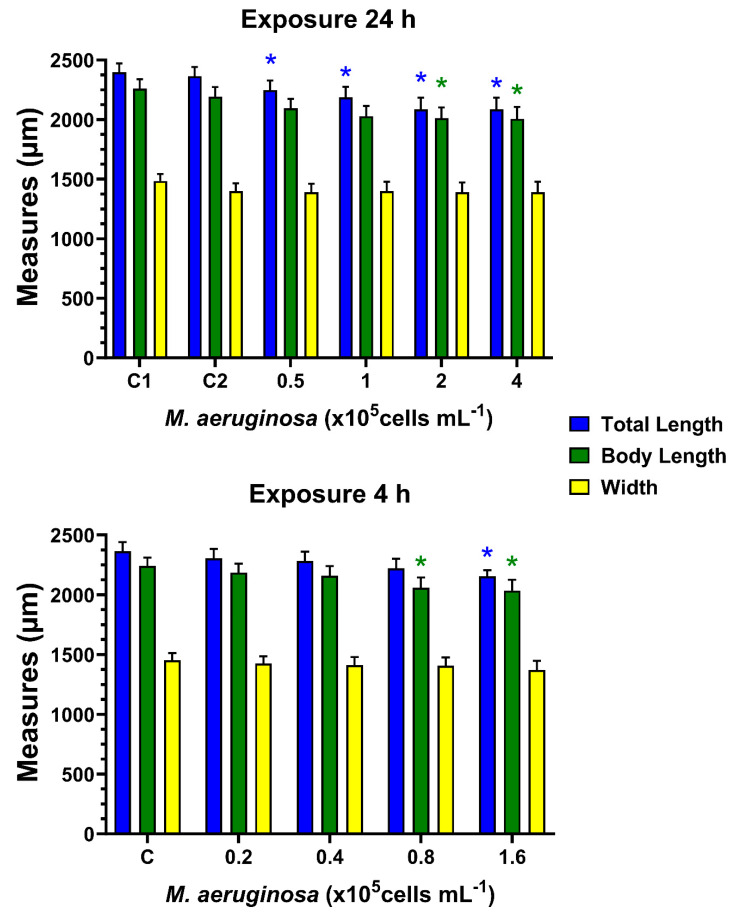
Total length, body length, and body width of *D. curvirostris* adults alternately exposed to the toxigenic cyanobacterium *M. aeruginosa* for 24 h and 24 h recovery, and 4 h with 44 h recovery. Bars represent mean values ± standard error. Asterisks indicate significant differences compared to the control group for each measure (Dunnett’s test, *p* < 0.05).

**Figure 6 toxins-16-00360-f006:**
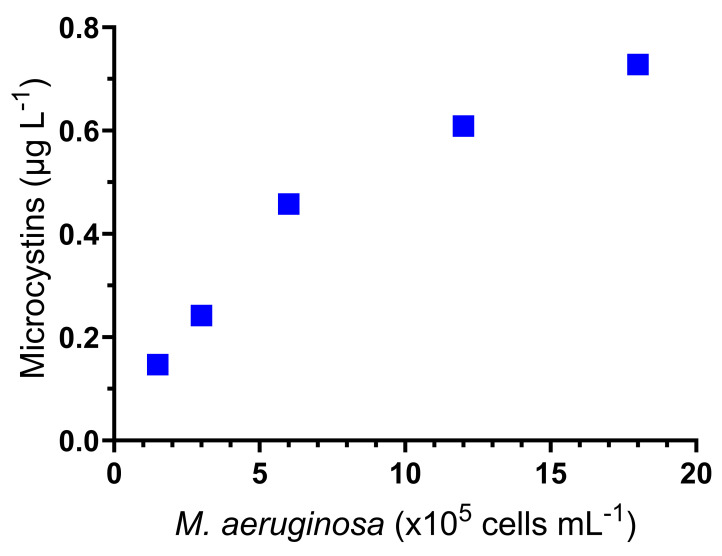
The concentration of microcystins (expressed as microcystin-LR), recorded in the different cell densities of *M. aeruginosa* used to determine the LC_50_ in *D. curvirostris*.

## Data Availability

The data presented in this study are available on request from the corresponding author.

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
