# Peer review of "Continuous and Intermittent Exposure to the Toxigenic Cyanobacterium Microcystis aeruginosa Differentially Affects the Survival and Reproduction of Daphnia curvirostris"

_toxins, 2024, doi:10.3390/toxins16080360_

Round 1
Reviewer 1 Report
Comments and Suggestions for Authors
Review of toxins-3114297: Continuous and intermittent exposure to the toxigenic cyanobacterium Microcystis aeruginosa differentially affects the survival and reproduction of Daphnia curvirostris
The authors demonstrate the effect of cell concentration, cell extract and filtrate of Microcystis aeruginosa on some life history parameters in Daphnia curvirostris. This is quite extensive study, composed of several experiments and with enough number of replicates, although it is based on only one Daphnia clone and one toxic strain of Microcystis. The authors found that continuous exposure to Microcystis cells had the strongest negative effects on Daphnia performance. In my opinion, the manuscript needs substantial revision and improvements. It is a pity that line numbers are not added to the manuscript.
Some phrases and words in the manuscript are not adequately used and need correction: e.g. “Eutrophication produces habitat degradation…” – eutrophication leads to habitat degradation; “Continuous exposure to Microcystis cells produced 100% mortality…” – it caused or triggered 100% mortality…; the word “episodic” refers to something that occurrs occasionaly and at irregular intervals, while in one of the experiments you had regular periods, in which daphnids experienced M. aerugionosa. Consider replacing “Massive growths…” by “Massive occurrence…”. Blooms usually occurs in late summer – there are also some exemplary lakes which experience cyanobacterial blooms throughout the year, also in wintertime. “Microcystis aeruginosa is the most important and frequent…” – “…is one of the most frequent…”. “mucilage masses” – please consider using “sheaths” instead of masses. “…in the planktonic structure of freshwater systems…” – consider replacing this part of sentence by “…for the functioning of freshwater ecosystems”. “Cladocerans are non-selective filter feeder…” – we can find predators and scrapers among the members belonging to this group. “…which include their ability to locate at different depths in the water column (due to the presence of aerotopes) – please consider emphasizing surface blooms – thanks to aerotopes cyanobacteria can maintain positive buoyancy, and hence some bloom-forming species have tendency to form surface blooms, which i.e. lower competitive abilities among other phytoplankton organisms, lower a chance for macrophyte development etc. “… develop in short times and…” – please be more precise what does short time exactly mean; “Exposure of D. curvirostris to the toxigenic strain of M. aeruginosa VU5 produced toxic…” – this sentence is not clear, the word “produced” is inadequate here.
responses that ranged; “three concentrations (4, 8, and 12 x10⁵ cellsmL-1) were fed and…” – how concentrations can be fed?
Although the backround is in general well-provided in the Introduction, aims are well defined, the authors did not formulate any hypothesis to be tested by the experiments.
Material and Methods: Description of your model organisms is insufficient. Please provide details about the clone of D. curvirostris you used in the study, conditions at which stock cultures were grown (temp, photoperiod, etc.). What was the origin of the clone? Whether this clone was isolated from the lake where Microcystis was present, where microcystins were present? Can we suspect that this clone migth have experience with Microcystis and MCs in the nature? How long the clone is under lab cultivation? How the body size was determined in daphnids?
What about other toxic metabolites in VU-5 strain of Microcystis? Do you have any information if this strain produces other secondary metabolites that might affect Daphnia performance?
Microcystis cell concentrations: why the authors used different concentrations of cells in particular experiments? (Acute toxicity: 1.5, 3, 6, 12, 18x10^5 cells/ml; Bioassay with continuous exposure to Microcystis: only 4, 8, 12x10^5 cells/ml; Intermittent exposure bioassay: 0.5, 1, 2, 4x10^5 cells/ml). How the abundance of Microcystis cells was determined?
"In my opinion, the experimental design in 2.3.1 and 2.3.2 is inapriopriate. Microalga P. subscapitata at a concentration of 1x10
6 cells/ml should also be present in the treatments with Microcystis treatments. Considering the microalga absence in the treatments with Microcystis, the control used by the authors is not the true control, this is the treatment with another type of food. Thus, I am not sure if Dunnett'test is a proper post-hoc test here. I would suggest conducting multiple-pairwise comparisons between all groups, e.g. Tukey's test.
Quantification of microcystins: ELISA is a rapid test that allow to detect toxic compounds. Application of chromatographic techniques would be very useful and allow on precise detection of MCs variants present in this strain.
Results: Statistical analyses are very poorly presented – e.g. there is no results of ANOVA tests for life history traits of Daphnia.
Figure 6: In my opinion, expressing microcystins concentrations as microcystin-LR based on ELISA is inapriopriate. The ELISA immunoassay does not distinguish between the microcystin variants.
Discussion:
“However, other studies (Pawlik-Skowrońska et al., 2019) report that the…” – you provided single citation, so it should be“…other study (Paw...) reports…”.
“This situation did not occur in our study since the most significant toxic effects were recorded in the exposure…” - Weaker reduction of D. curvirostris mortality in the presence of ACE perhaps could be the result of degradation of other sec. metabolites during preparation of the extract (e.g. heating). Do you have information about other toxins that this strain of Microcystis can produce? If yes, you could check the stability of these toxins in response to heating simply from the literature. MCs are stable and resistant to heating but what about other secondary metabolites that this strain may produce?
Penultimate parapgraph: Why there is a big difference in microcystins concentrations between ACE (2.1 ug/L) and cells (1.6 ug/L) since CFM was free of microcystins? Considering the applied experimental design in this study, what other factors could affect the observed life history responses of Daphnia to Microcystis cells? This problem should be discussed in sufficient details.
Reviewer 2 Report
Comments and Suggestions for Authors
The topic is interesting to understand how toxigenic cyanobacterium affect the survival and reproduction of crustaceans. However, to proceed with the manuscript, further clarifications and corrections are needed.
1. The manuscript needs to undergo extensive English editing for grammar and English.
2. The absence of line numbering in the provided manuscript makes it challenging for reviewers to provide specific comments.
3. In page 1, line-6 "cells-free culture medium (CFM)" should be written as "cell-free culture medium (CFM)".
4. "In-text citations should be in numerical format.
5. On page 13, line 40, please write out the full abbreviation of MCs.
6. On page 3, line 32, please clearly define abbreviation of OECD test method.
7. The referencing style should adhere to journal format.
Comments on the Quality of English LanguageThe manuscript needs to undergo extensive English editing for grammar and English.
Round 2
Reviewer 1 Report
Comments and Suggestions for Authors
Review of toxins-3114297: Continuous and intermittent exposure to the toxigenic cyanobacterium Microcystis aeruginosa differentially affects the survival and reproduction of Daphnia curvirostris
The authors improved the manuscript and now it looks much better for me. Statistical analyses are provided in more details. I appreciate that the authors detailed the characteristics of their model organisms and decided to conduct multiple pairwise comparisons by Tukey’s. Additional explanations provided by the authors about the design of the life history experiments clarified this issue for me and thus I do not have more criticism on this.
I have only minor remarks to the revised manuscript.
Page 3, line 99-101: Please consider starting the sentence like that: “In this study, we tested whether…”.
Page 2, line 70: Please replace the word “planktonic” by “planktic. I am aware that this word is in use in scientific literature, but this derivation is incorrect. You can find explanation in the following paper:
Emiliani C. Planktic/planktonic, nektic/nektonic, benthic/benthonic. Journal of Paleontology. 1991;65(2):329-329. doi:10.1017/S0022336000020576
Page 4, line 160: Please consider replacing “26 d” by “26 days”.
